# The Risk of Pyelonephritis Following Uncomplicated Cystitis: A Nationwide Primary Healthcare Study

**DOI:** 10.3390/antibiotics11121695

**Published:** 2022-11-24

**Authors:** Filip Jansåker, Xinjun Li, Ingvild Vik, Niels Frimodt-Møller, Jenny Dahl Knudsen, Kristina Sundquist

**Affiliations:** 1Center for Primary Health Care Research, Department of Clinical Sciences Malmö, Lund University, 205 02 Malmö, Sweden; 2Department of Clinical Microbiology, Rigshospitalet, DK-2100 Copenhagen, Denmark; 3The Antibiotic Centre for Primary Care, Department of General Practice, Institute of Health and Society, University of Oslo, 0318 Oslo, Norway; 4Oslo Accident and Emergency Outpatient Clinic, City of Oslo Health Agency, 0661 Oslo, Norway; 5Center for Community-Based Healthcare Research and Education (CoHRE), Department of Functional Pathology, School of Medicine, Shimane University, Shimane 693-8501, Japan; 6Department of Family Medicine and Community Health, Department of Population Health Science and Policy, Icahn School of Medicine at Mount Sinai, New York, NY 10029, USA

**Keywords:** antibiotics, cervical cancer, complications, cystitis, parity, pyelonephritis, sociodemographic factors, treatment

## Abstract

Background: The risk of pyelonephritis following uncomplicated lower urinary tract infection (cystitis) in women has not been studied in well-powered samples. This is likely due to the previous lack of nationwide primary healthcare data. We aimed to examine the risks of pyelonephritis following cystitis in women and explore if antibiotic treatment, cervical cancer, parity, and sociodemographic factors are related to these risks. Methods: This was a nationwide cohort study (2006–2018) of 752,289 women diagnosed with uncomplicated cystitis in primary healthcare settings. Of these, 404 696 did not redeem an antibiotic prescription within five days from cystitis. Logistic regression models were used to calculate odds ratios for pyelonephritis within 30 days and 90 days following the cystitis event. Results: Around one percent (7454) of all women with cystitis were diagnosed with pyelonephritis within 30 days, of which 78.2% had not redeemed an antibiotic for their cystitis. Antibiotic treatment was inversely associated with both outpatient registration and hospitalization due to pyelonephritis, with odds ratios of 0.85 (95% CI 0.80 to 0.91) and 0.65 (95% CI 0.55 to 0.77), respectively. Sociodemographic factors, parity, and cervical cancer were, with few exceptions (e.g., age and region of residency), not associated with pyelonephritis. Conclusions: Antibiotic treatment was inversely associated with pyelonephritis, but the absolute risk reduction was low. Non-antibiotic treatment for cystitis might be a safe option for most women. Future studies identifying the women at the highest risks will help clinicians in their decision making when treating cystitis, while keeping the ecological costs of antibiotics in mind.

## 1. Introduction

Uncomplicated lower urinary tract infection (cystitis) is one of the most common community-acquired infections in women and is caused by bacteria (mainly *Escherichia coli*). The most common symptoms are dysuria, urgency, and pollakiuria [1,2]. About 10% of women have at least one cystitis each year, and over 60% experience it at least once during their lifetime [3], with sociodemographic variations in incidence rates [1]. On the other hand, upper urinary tract infection (acute pyelonephritis) is a much more serious infection that mostly occurs in the same group of otherwise healthy women who are at risk of cystitis [3]. The incidence rates are about 20-fold lower [3,4] and with similar sociodemographic differences [4]. However, while pyelonephritis requires antibiotic therapy [3,5,6], cystitis might not [2,7,8].

Cystitis is considered to be a self-limiting condition in many cases. For example, placebo-controlled studies found that about one in four patients with cystitis treated with placebo were cured, without any evidently relevant risk difference for pyelonephritis compared to those treated with antibiotics [8,9,10]. We recently found that little over half the women diagnosed with cystitis redeemed an antibiotic prescription from a pharmacy in Sweden, with minor sociodemographic variations in antibiotic treatment rates [11,12].

The current recommendation for uncomplicated cystitis from 2017 by the Swedish Medical Products Agency is watchful waiting if the symptoms are not considered severe for the patient; mild symptoms should be managed without antibiotics [13]. Although suggestions on similar strategies have appeared elsewhere [2], few other national guidelines for cystitis have, to our knowledge, aligned with this recommendation; this is perhaps due to the opinion that antibiotic therapy might actually lower the risk of complications such as pyelonephritis—but the evidence is inconclusive.

Although many women with cystitis might require antibiotic therapy, a large group could manage without it [8,10,14,15,16]. This has been tempting to implement considering the rising threat of antibiotic resistance, and such a recommendation would be likely to lower antibiotic use and hence reduce the development of antibiotic resistance [17,18,19,20]. However, before a general recommendation on this can be made, sufficiently powered large studies on the risk of pyelonephritis in patients treated with antibiotics or not are required. As the risk seems to be quite low, such high-powered trials are rarely possible in randomized controlled trial settings. Instead, population-based studies on data from primary healthcare settings, where the majority of cystitis cases are managed and treated, could add more evidence to previous clinical trials. Hitherto, such studies have not been possible, likely due to the previous lack of large-scale population-based data from primary healthcare clinics.

This study aimed to estimate the risk of pyelonephritis following a cystitis event in women and explore if the risk was affected by antibiotic treatment in general and the two most used first-line antibiotics [5] in Sweden (pivmecillinam and nitrofurantoin) [13] in particular. We also aimed to explore sociodemographic factors, parity, and cervical cancer in this context, to explore factors that potentially could affect the risk of pyelonephritis following cystitis.

## 2. Materials and Methods

### 2.1. Design and Setting

This was a nationwide open cohort register study of 752,289 women diagnosed with acute uncomplicated cystitis in primary healthcare settings. The women were aged 18–65 years at the time of inclusion and could only be included once. Baseline occurred when a woman was first diagnosed with cystitis during the study period. The study was conducted at Lund University, Sweden.

### 2.2. Study Population

The sampling criteria to identify the women with acute uncomplicated cystitis were identical to a previous study by our group [11], i.e., the first occurrence of an acute uncomplicated cystitis episode (hereafter referred to as cystitis) in a primary healthcare setting. The 10th revision of the International Classification of Diseases (ICD-10) diagnosis code N30 (but not N301, N302, N303, N304, and N308) was used to identify the cases of cystitis. Women with complicating factors more aligned with a diagnosis of complicated cystitis [3,21,22,23] were not eligible for inclusion: i.e., history of urological neoplasms; HIV or other immunodeficiency disorders; diabetes mellitus; paraplegic syndrome; chronic nephritic syndrome; hereditary or acquired nephropathy; chronic pyelonephritis; hydronephrosis; other serious kidney diseases; kidney failure; urolithiasis; neurological bladder dysfunction; and congenital or other diseases of the kidney, bladder, or urinary tract (ICD-10: B20-24, C64-68, D41, D80-89, E10-11, M623, N03, N07, N11, N13-23, N25-29, N32, Q60-64). Women with a history of redeemed prescriptions for anti-neoplastic and/or immunomodulating agents or corticosteroids for systemic use were also not considered. Women with pyelonephritis (outcome) prior to the redeemed antibiotic prescription (exposure) were excluded from the logistic regression analyses.

### 2.3. Study Period

Women were included if they had an episode of cystitis in the period spanning from 1 January 2006 to the 2 October 2018. Each woman could only be included once in the study. The follow-up period for pyelonephritis ended on 31 December 2018 (i.e., 90 days after the last day of inclusion for cystitis).

### 2.4. Main Exposure

The main exposure considered was oral antibiotic treatment or not (reference). Antibiotic treatment was defined as any class of oral antibiotic generally used for or recommended for urinary tract infections (UTIs) [11,12]. Antibiotic treatment was defined as the first redeemed antibiotic treatment within five days (or two days in the sensitivity analysis) from the cystitis diagnosis. The following antibiotic groups were considered (ATC code): penicillins with extended spectrum (J01CA); nitrofuran derivatives (J01XE); trimethoprim and derivatives (J01EA); fluoroquinolones (J01MA); cephalosporins (J01DB/J01DE/J01DI); others (J01MB, other quinolones; J01EB/J01EC/J01ED, sulfonamides (less than 10 cases); J01EE, sulfonamides and trimethoprim combinations; and J01CR, amoxicillin and clavulanic acid). During 2006–2013, when seven levels on the ATC codes were available to us, the antibiotic groups of J01CA, J01XE, J01EA, and J01MA redeemed for uncomplicated cystitis were more or less exclusively pivmecillinam (J01CA08), nitrofurantoin (J01XE01), trimethoprim (J01EA01), ciprofloxacin (J01MA02), and norfloxacin (J01MA02) [12].

### 2.5. Covariates

Since sociodemographic factors (age, educational level, family income level, region of residence, and country of origin), parity, and cervical cancer seem to be related to cystitis [1], antibiotic treatment for cystitis [11], and pyelonephritis [4], we included these covariates in the analysis.

### 2.6. Outcomes

Acute pyelonephritis was defined as an ICD-10 code of N10 or N12 within 30 and 90 days after the cystitis event. The first outcome measure was the occurrence of N10 or N12 in outpatient settings (i.e., primary healthcare data or the National Patient Register of outpatient specialist care data). The second outcome measure was hospitalization with a diagnosis of N10 or N12 in the National Patient Register of inpatient data.

### 2.7. Sources of Data

The study population was included using primary healthcare data (1997–2018) collected from 20 out of 21 administrative regions in Sweden. Data on complicating factors (exclusion criteria) were collected from primary healthcare data and the Swedish Prescribed Drug Register (2006–2018). The main predictor (antibiotic treatment) was identified in the Swedish Prescribed Drug Register. Sociodemographic factors were collected from the Total Population Register (1968–2018). The outcomes were identified using primary healthcare data, as well as outpatient (2001–2018) and inpatient data (1964–2018) from the National Patient Register (NPR). Parity data were collected from the National Medical Birth Register (1973–2018), and cervical cancer data were collected from the National Cancer Register. Data from the Total Population Register was provided by Statistics Sweden (Swedish: Statistiska centralbyrån, SCB), and the data from the other registers were provided by the National Board of Health and Welfare (In Swedish: Socialstyrelsen).

### 2.8. Statistics and Sampling of Study Population

The study period started on 1 January 2006 as an open cohort study, and women were followed from the age of 18 years (or immigration before 65 years of age) until death, emigration, 65 years of age, or end of the study period on 31 December 2018, whichever came first. Women were included in the study population at their first occurrence of cystitis in a primary healthcare setting. Covariates were collected at the time of the cystitis event. The main predictor, i.e., exposure to “antibiotic treatment” or “no antibiotic treatment” (reference) was measured within five days (exposure window) from the cystitis event. Women were followed for a total of 90 days for pyelonephritis; the endpoints were measured at 30 and 90 days. A flowchart was constructed on the sampling of the study population, exposure group sizes, and number of outcome events. Descriptive statistics were calculated for each covariate in relation to the hospitalization due to pyelonephritis and pyelonephritis treated in an outpatient setting (outpatient pyelonephritis).

The rates of pyelonephritis in women with cystitis were estimated in total and stratified by antibiotic treatment or not. In this crude analysis, women with pyelonephritis prior to a redeemed antibiotic prescription within the 5-day exposure window for antibiotic treatment were included in the “no treatment group”. Absolute risk differences and numbers needed to treat with 95% confidence intervals (CIs) were calculated for antibiotic treatment in general and for each of the two most redeemed [11] first-line antibiotics compared to no antibiotic treatment.

In the more comprehensive epidemiological analysis, we explored potential factors related to the progression of pyelonephritis following cystitis. In this analysis, logistic regression models were used to estimate odds ratios (ORs) and 95% CIs for any association between antibiotic treatment, the covariates, and the outcome (pyelonephritis). Three models were applied in the analysis: Model 1 was a univariate model, Model 2 was adjusted for age and sociodemographic factors, and Model 3 was adjusted for all covariates. These models were analyzed with the following two definitions of the outcome: (i) pyelonephritis in outpatient settings and (ii) hospitalization due to pyelonephritis within 30 days or 90 days from the cystitis event. In these models, women with pyelonephritis prior to antibiotic exposure within the 5-day exposure window were excluded from the statistical analysis. A sensitivity analysis on outpatient pyelonephritis was also conducted, in which the models were replicated with a 2-day exposure window and 7-day and 30-day follow-up periods for pyelonephritis following cystitis.

Missing values of the covariates (range 0.0–0.6%) were treated as follows: For education and income, missing values were included in the groups with the lowest levels of education and income. Unknown region of residence and unknown country of origin (in total, 40 individuals) were included in the category “large cities” and the category “Sweden”, respectively. Kaplan–Meier curves were also plotted. A two-tailed *p*-value of <0.05 was considered for statistical significance. SAS version 9.4 (SAS Institute Inc.; Cary, NC, USA) was used for all statistical analyses.

## 3. Results

Appendix A shows the flowchart of the 752,289 women with uncomplicated cystitis. The rates of outpatient pyelonephritis and hospitalization due to pyelonephritis within 30 days of the cystitis event were 0.99% and 0.34%, respectively.

Table 1 (and Appendix A) includes individual-level characteristics of the 748,704 women followed for outpatient pyelonephritis. The table also shows that 45.9% of women had redeemed an antibiotic prescription within 5 days following their first cystitis diagnosis in the time period. Of these antibiotics, pivmecillinam (J01CA) accounted for 58.5%, followed by nitrofurantoin (J01XE) and trimethoprim (J01EA), which accounted for 24.8% and 10.3%, respectively. For pivmecillinam and nitrofurantoin, the 30-day absolute risk of pyelonephritis was 0.24% and 0.21%, respectively, compared to 1.43% for women without antibiotic treatment. Of all women with pyelonephritis following cystitis diagnosis, about 78.2% of all women with pyelonephritis following their cystitis event did not redeem an antibiotic.

Table 2 shows the risk differences between women treated with antibiotics and women not treated with antibiotics. It also includes estimations of the number of women with uncomplicated cystitis that need to be treated with antibiotics to save one woman from pyelonephritis.

Table 3 includes the fully adjusted models on the 30-day and 90-day risk of outpatient pyelonephritis following a cystitis event in relation to antibiotic treatment or not. Antibiotic treatment was inversely associated with pyelonephritis in general, but only the three most common groups of antibiotics were inversely associated with pyelonephritis in the stratified analysis. For example, the most used antibiotic (pivmecillinam) was associated with an OR of 0.44 (95% CI 0.40 to 0.48) for pyelonephritis within 30 days from the cystitis event.

Table 4 includes a fully adjusted model from Appendix A. This table shows that sociodemographic factors, history of cervical cancer, and parity were mostly not associated with pyelonephritis in relation to antibiotic treatment or not. However, high age, low income, and originating from the Middle East were inversely associated with pyelonephritis compared to their corresponding references, while living in northern Sweden was associated with higher odds of pyelonephritis. Furthermore, as can be seen in Appendix A, the association between antibiotic treatment for cystitis and pyelonephritis remained after adjusting for these covariates. The results also remained significant and were similar in the sensitivity analyses (Appendix A). Appendix A include Kaplan–Meier plots for time to pyelonephritis.

Table 5 shows that antibiotic treatment for uncomplicated cystitis was inversely associated with hospitalization due to pyelonephritis in a similar manner to that of outpatient pyelonephritis, but the rates for hospitalization due to pyelonephritis were lower (Appendix A). Sociodemographic factors, history of cervical cancer, and parity were generally not associated with hospitalization due to pyelonephritis (Appendix A), but high age and living in Southern Sweden were inversely associated with hospitalization due to pyelonephritis compared to their corresponding reference group.

## 4. Discussion

This study is, to our knowledge, the first nationwide study exploring the risk of acute pyelonephritis following an uncomplicated cystitis event in women. The main findings were that around one percent of all women diagnosed with cystitis were diagnosed with pyelonephritis within 30 days from the cystitis event, of which over half seemed to acquire it in proximity (within the first five days) to the cystitis event. The rate of hospitalizations due to pyelonephritis was about one-third of the outpatient pyelonephritis rate. Antibiotic treatment for cystitis was, in general, inversely associated with pyelonephritis, particularly for the two most used antibiotics: pivmecillinam and nitrofurantoin. However, the absolute risk differences between antibiotic treatment and no treatment were quite low.

Clinical studies have been inconclusive in their findings regarding the risk of pyelonephritis in women with cystitis. Previous randomized controlled studies comparing placebo to antibiotic therapy for cystitis have, to our knowledge, not shown any increased risk of pyelonephritis in the placebo-treated groups [8,9,16]. For example, in a large placebo-controlled trial [8] with 1143 women (288 treated with placebo), there were only two cases of pyelonephritis: one in each of the placebo and pivmecillinam treatment arms. An earlier placebo-controlled trial with 78 women (38 treated with placebo) found only one case of pyelonephritis in a woman treated with placebo. However, a pivmecillinam-only trial by our group [24] found that about 2% of the women with cystitis developed pyelonephritis during the follow-up period. Considering this together with the findings of the present study, it is likely that the inconclusive results on placebo and antibiotics are due to the infeasibility of performing sufficiently powered clinical studies exploring the risk of pyelonephritis in women with cystitis.

The main mechanism behind the inverse association between antibiotic treatment for cystitis and pyelonephritis is likely the antimicrobial (e.g., bacterial killing) effect [25] on the pathogens causing most cases of cystitis [3,5,8,9,10,14,24]. Thus, antibiotic treatment for cystitis lowers the probability of the bacteria ascending the urinary tract and causing pyelonephritis. This would explain the general findings and the findings related to the first-line treatment options of nitrofurantoin and pivmecillinam (83.4% of all antibiotic treatments). On the other hand, the contradictive findings concerning the small group (3%) of women with cystitis treated with fluoroquinolones (2.3%), cephalosporins (0.6%), or other antibiotics (0.1%) most likely have other explanations. One explanation might be the relatively high resistance rates for some of these antibiotics in Sweden [26]. Another contributing factor may be that most of these antibiotics mainly have been recommended for outpatient treatments of pyelonephritis. Fluoroquinolones are recommended as first-line antibiotics for outpatient pyelonephritis in both Sweden and internationally [5,13]. Thus, it is possible that these women already had pyelonephritis symptoms at the time of inclusion and were mislabeled with a cystitis diagnosis but prescribed the recommended outpatient treatment for pyelonephritis.

The main limitation of epidemiological studies of this caliber is the lack of data on symptoms and clinical presentation. Therefore, we could not assess if the cystitis diagnoses were correct or if some of the patients were misdiagnosed. For example, some patients could have had asymptomatic bacteriuria mislabeled as cystitis diagnosis, but (correctly) not treated with antibiotics, or prescribed an unnecessary antibiotic treatment. It is also possible that some patients with early onset of pyelonephritis following a cystitis diagnosis actually had symptoms of pyelonephritis at the time of the cystitis diagnosis being registered. Nor was it possible to evaluate whether antibiotic treatment or non-treatment depended on differences in the patients’ symptoms. Secondly, we did not have access to microbiological data and can therefore not assess if antibiotic treatment failures were due to resistant bacteria or if the increased risk of pyelonephritis in women without antibiotic treatment was associated with the presence of concomitant bacteriuria. For example, it is plausible that women without bacteriuria suffer a much lower risk of pyelonephritis if not treated with antibiotics for their cystitis event [14]. Further studies using microbiological data are needed to explore this. Thirdly, our study did not have access to the proportion of antibiotic prescriptions that were not redeemed. Therefore, no conclusion can be drawn on whether no antibiotic treatment was due to no prescription (healthcare) or lack of adherence (patient). However, it has been demonstrated in the USA [27] that nearly all antibiotic prescriptions seem to be redeemed by the patients. Fourthly, due to the design of the study, the analyses did also not take the number of infections (i.e., recurrent infections) into account as each woman was only set to be included once in the study; thus, we were not able to assess if recurrent infections were a confounding variable. Finally, we did not have access to information on non-antibiotic treatments (e.g., increased fluid intake) for the cystitis event or information on whether the patient used antibiotics bought abroad or leftover antibiotics from another cystitis episode. Nevertheless, considering the comprehensive and extensive nature of our data, the limitations were likely balanced by the strengths.

The major strengths of this study were that we were able to assess the risk of pyelonephritis in a very large (nationwide) cohort of women diagnosed with uncomplicated cystitis in primary healthcare settings and that the study involved several validated nationwide data registries, thus enabling us to fill in the gaps that clinical trials hitherto have been unable to fully address: estimating how many women acquire pyelonephritis following cystitis and exploring the potential associations with antibiotic treatment and other factors. This is something which has not been possible to address in clinical trials due to the relative rareness of this complication. Furthermore, the proportions of specific antibiotics redeemed for cystitis were similar to the general use of these antibiotics in women during the same time period [26], and the rates of pyelonephritis were in the same order of magnitude implied from randomized controlled trials [8,9,14,15,16,24,28,29]. These consistencies support that the data sources are representative and can be used to identify most urinary tract infections and antibiotic treatments.

The findings support the current guidelines that do not recommend immediate antibiotic treatment for women with mild–moderate symptoms of cystitis [13]. Our findings—together with previous clinical studies [3,8,9,10,15]—imply that most women with uncomplicated cystitis can be safely managed without antibiotic treatment and that antibiotic treatment for women with cystitis should perhaps, in most cases, be given for other reasons than that of preventing pyelonephritis. Lowering the use of antibiotics for this very common infection could save otherwise healthy women from potentially unnecessary antibiotic exposure. This would not only benefit the individual patient but also society as a whole, considering the ecological, health, and financial costs attributed to the increased antibiotic resistance worldwide [17,20]. However, further research is needed to identify risk factors for pyelonephritis in patients with cystitis, especially when antibiotics cannot be omitted, as this study suggests that over four out of five women who suffer pyelonephritis progression following a cystitis event were not treated with antibiotics. Nevertheless, in waiting for such data, we suggest, based on previous evidence [8,9,10,14,15,16] together with our findings, that otherwise healthy women with mild symptoms can be advised to manage without antibiotics. On the other hand, if, for example, the woman is severely bothered by her symptoms and has a positive urine dipstick [7,30], first-line antibiotic treatment for cystitis with low rates of resistance in the community [5] could be prescribed, perhaps with a recommendation to wait and see in a shared decision-making process [2].

Future prospective clinical studies are needed to better understand which women suffer a risk of pyelonephritis if not treated with antibiotics for cystitis. These could be used not only to confirm our findings, but also to characterize what factors are behind the risk of developing pyelonephritis when not treated with antibiotics for cystitis. This would help clinicians and guideline committees to provide better individualized treatment and safer guidelines for women worldwide suffering from this very common generally self-limited infective condition. We have recently identified that factors likely beyond the infection (e.g., sociodemographic factors) seem to affect the risk and treatment of cystitis [1,11,12] as well as the risk of pyelonephritis [4]. With few exceptions (e.g., age), sociodemographic factors did not seem to be associated with pyelonephritis in this present study, indicating that sociodemographic inequalities of healthcare are likely not related to increased risk of pyelonephritis in women with cystitis. Instead, other factors in the host (e.g., urological comorbidities or immunosuppressant mechanism) and the pathogen (e.g., virulence factors, natural or acquired resistance) might include the causal mechanisms behind the risk difference of pyelonephritis in women treated or not treated with antibiotics for symptomatic cystitis in primary healthcare. In particular, factors related to the early progression of pyelonephritis due to cystitis need to be more extensively studied, as these women are probably more likely to benefit the most from early/immediate treatment with antibiotics. The results from the sensitivity analysis may indicate such a possibility. In the sensitivity analysis (Appendix A), any antibiotic treatment redeemed within two days from the cystitis event was associated with an OR of 0.71 (95% CI 0.66 to 0.76) for outpatient pyelonephritis within 30 days (compared to no antibiotic treatment). In the main analysis (Table 4), the corresponding OR associated with any antibiotic treatment redeemed within 5 days from the cystitis event was 0.85 (95% CI 0.80 to 0.91).

## 5. Conclusions

In summary, this study suggests that women with uncomplicated cystitis suffer a significantly higher risk of pyelonephritis if not treated with antibiotics. However, although the nationwide data provided statistically significant results, the clinical significance for the individual woman should be minor. Future studies identifying the women at the highest risks will help clinicians in their decision making when treating cystitis, while keeping the ecological costs of antibiotics in mind.

## Figures and Tables

**Table 1 antibiotics-11-01695-t001:** The study population of women diagnosed with cystitis and number of cases diagnosed with pyelonephritis within 30 days and within 90 days in relation to antibiotic treatment.

	Total Population		With Acute Pyelonephritis within:					
			30 Days			90 Days		
	No.	Proportion %	No.	Proportion %	Case Rate %	No.	Proportion %	Case Rate %
Total Study Population Descriptive data on all women with/without antibiotic treatment and cases of acute pyelonephritis								
No antibiotic treatment	408,281	54.3	5829	78.2	1.43	6574	77.6	1.61
Any antibiotic ^1^	344,008	45.7	1625	21.8	0.47	1898	22.4	0.55
All	752,289	100.0	7454	100.0	0.99	8472	100.0	1.13
Individuals included in the Logistic Regression Model (Excluding 3585 women with pyelonephritis prior to antibiotic treatment)								
No antibiotic treatment ^2^ (in analysis)	404,696	54.1	2244	58.0	0.55	2989	61.2	0.74
Any antibiotic ^1^	344,008	45.9	1625	42.0	0.47	1898	38.8	0.55
Penicillins with extended spectrum (J01CA) ^3^	201,114	26.9	490	12.7	0.24	643	13.2	0.32
Nitrofuran derivatives (J01XE) ^3^	85,238	11.4	177	4.6	0.21	235	4.8	0.28
Trimethoprim and derivatives (J01EA)	35,361	4.7	51	1.3	0.14	63	1.3	0.18
Fluoroquinolones (J01MA)	17,079	2.3	828	21.4	4.85	868	17.8	5.08
Cephalosporins (J01DB-E,I)	4136	0.6	37	1.0	0.89	44	0.9	1.06
Others ^4^	1080	0.1	42	1.1	3.89	45	0.9	4.17
All	748,704	100.0	3869	100.0	0.52	4887	100.0	0.65

Appendix A includes all descriptive data on the study population. ^1^ Redeemed antibiotic prescription within five days from the cystitis event and before the pyelonephritis event; ^2^ 3585 women were excluded due to diagnosis of outcomes before antibiotic treatment; ^3^ more or less only pivmecillinam and nitrofurantoin; ^4^ others included ATC codes J01MB, J01EB, J01EC, J01ED, J01EE, and J01CR. Nationwide primary healthcare data were used to identify the study population of women with uncomplicated cystitis. The Swedish Prescribed Drug Register was used to identify antibiotic treatment. Nationwide primary healthcare data and the National Patient Register (outpatient data) were used to identify pyelonephritis.

**Table 2 antibiotics-11-01695-t002:** The 30-day crude risk of acute outpatient pyelonephritis in women with acute uncomplicated cystitis.

	All Cases	Antibiotic Treatment	No Treatment (*n* = 408,281)	Absolute Risk Reduction % (Number Needed to Treat)		
	No. (%)	No. (%)	No. (%)	ARR % (NNT)	95% CI	
Any antibiotic (*n* = 344,008)	7454 (0.99)	1625 (0.47)	5829 (1.43)	0.95 (105)	0.91 (110)	1.00 (100)
Pivmecillinam ^1^ (*n* = 201,114)	7454 (0.99)	490 (0.24)	5829 (1.43)	1.19 (85)	1.16 (88)	1.22 (82)
Nitrofurantoin ^1^ (*n* = 85,238)	7454 (0.99)	177 (0.21)	5829 (1.43)	1.22 (82)	1.16 (85)	1.28 (79)

^1^ Pivmecillinam (J01CA) and nitrofurantoin (J01XE). ARR: Absolute risk reduction. CI: Confidence interval. NNT: number needed to treat. No.: number. Nationwide primary healthcare data were used to identify the study population of women with uncomplicated cystitis: 752,289. The 3858 cases with pyelonephritis prior to outcome in the exposure period of five days were included in the “no treatment” group in this analysis. Antibiotic treatment was identified in the Swedish Prescribed Drug Register. Outpatient pyelonephritis was identified in nationwide primary healthcare data and the National Patient Register (outpatient data).

**Table 3 antibiotics-11-01695-t003:** The association between antibiotic treatment (redeemed within five days) for acute uncomplicated cystitis and acute outpatient pyelonephritis within 30 and 90 days from the cystitis event (2006–2018).

	Pyelonephritis within 30 Days				Pyelonephritis within 90 Days			
Treatment Groups	OR ^1^	95% CI		*p*-Value	OR ^1^	95% CI		*p*-Value
All antibiotic groups (ref. Non)	0.85	0.80	0.91	<0.0001	0.75	0.70	0.79	<0.0001
Specific antibiotic groups (ref. Non)								
Penicillins with extended spectrum (J01CA)	0.44	0.40	0.48	<0.0001	0.43	0.39	0.47	<0.0001
Nitrofuran derivatives (J01XE)	0.37	0.32	0.43	<0.0001	0.37	0.32	0.42	<0.0001
Trimethoprim and derivatives (J01EA)	0.26	0.20	0.35	<0.0001	0.24	0.19	0.31	<0.0001
Fluoroquinolones (J01MA)	9.32	8.59	10.12	<0.0001	7.34	6.79	7.94	<0.0001
Cephalosporins (J01DB-E,I)	1.66	1.20	2.30	0.0024	1.48	1.10	2.00	0.0102
Others ^2^	7.37	5.40	10.06	<0.0001	5.92	4.38	8.00	<0.0001

CI: confidence interval. OR: odds ratio. Nationwide primary healthcare data were used to identify the study population of women with uncomplicated cystitis. The Swedish Prescribed Drug Register was used to identify antibiotic treatment groups. Nationwide primary healthcare data and the National Patient Register (outpatient data) were used to identify pyelonephritis. ^1^ Fully adjusted for history of cervical cancer and parity and individual sociodemographic factors. ^2^ Others included J01MB, J01EB, J01EC, J01ED, J01EE, and J01CR (0.1% in total).

**Table 4 antibiotics-11-01695-t004:** The association between antibiotic treatment (redeemed within five days) for acute uncomplicated cystitis and acute outpatient pyelonephritis diagnosed within 30 days and 90 days from the cystitis event, adjusted for sociodemographic factors, cervical cancer, and parity (2006–2018).

	Pyelonephritis within 30 Days				Pyelonephritis within 90 Days			
Covariates	OR ^1^	95% CI		*p*-Value	OR ^1^	95% CI		*p*-Value
Antibiotic (ref. Non)	0.85	0.80	0.91	<0.0001	0.75	0.70	0.79	<0.0001
Age (ref. age 18–24 years)								
25–34	0.91	0.82	1.02	0.0930	0.87	0.79	0.95	0.0027
34–44	0.95	0.85	1.07	0.3927	0.85	0.76	0.94	0.0014
45–65	0.93	0.84	1.03	0.1393	0.87	0.80	0.95	0.0028
Educational level (ref. > 12 years)	1.06	0.99	1.13	0.1172	1.06	1.00	1.12	0.0708
Family income (ref. High)								
Low	0.88	0.80	0.98	0.0136	0.87	0.79	0.95	0.0015
Middle	0.96	0.89	1.04	0.2997	0.94	0.88	1.01	0.0943
Region of residence (ref. Large cities)								
Southern Sweden	0.97	0.90	1.05	0.4207	0.98	0.92	1.05	0.6312
Northern Sweden	1.17	1.05	1.29	0.0034	1.11	1.01	1.22	0.0243
Country of origin (ref. Born in Sweden)								
Eastern Europe	1.02	0.88	1.18	0.8233	1.00	0.88	1.15	0.9522
Western countries	0.95	0.81	1.12	0.5648	0.94	0.81	1.09	0.4242
Middle East/North Africa	0.78	0.67	0.92	0.0031	0.88	0.77	1.01	0.0788
Africa (excluding North Africa)	0.89	0.68	1.17	0.4045	0.82	0.64	1.06	0.1333
Asia (excluding Middle East) and Oceania	1.08	0.89	1.31	0.4532	1.10	0.92	1.30	0.2968
Latin America and the Caribbean	1.10	0.85	1.42	0.4905	1.07	0.85	1.36	0.5531
Cervical cancer (ref. Non)	0.98	0.91	1.06	0.5686	0.97	0.91	1.04	0.4532
Parity (ref. Non)	1.14	0.95	1.37	0.1479	1.11	0.95	1.31	0.1978

CI: confidence interval. OR: odds ratio. Nationwide primary healthcare data were used to identify the study population of women with uncomplicated cystitis. The Swedish Prescribed Drug Register was used to identify antibiotic treatment. Nationwide primary healthcare data and the National Patient Register (outpatient data) were used to identify pyelonephritis. ^1^ Fully adjusted for history of cervical cancer and parity and individual sociodemographic factors.

**Table 5 antibiotics-11-01695-t005:** The association between antibiotic treatment (redeemed within five days) for acute uncomplicated cystitis and hospitalization due to acute pyelonephritis within 30 days and 90 days from the cystitis event (2006–2018).

	Pyelonephritis within 30 Days				Pyelonephritis within 90 Days			
Treatment Groups	OR ^1^	95% CI		*p*-Value	OR ^1^	95% CI		*p*-Value
All antibiotic groups (ref. No treatment)	0.65	0.55	0.77	<0.0001	0.61	0.52	0.70	<0.0001
Specific antibiotic groups (ref. No treatment)								
Penicillins with extended spectrum (J01CA)	0.46	0.37	0.58	<0.0001	0.45	0.37	0.55	<0.0001
Nitrofuran derivatives (J01XE)	0.30	0.20	0.45	<0.0001	0.34	0.25	0.48	<0.0001
Trimethoprim and derivatives (J01EA)	0.43	0.26	0.72	0.0014	0.34	0.21	0.56	<0.0001
Fluoroquinolones (J01MA)	5.02	3.95	6.39	<0.0001	4.30	3.45	5.36	<0.0001
Cephalosporins ^2^ (J01DB-E,I)	1.72	0.81	3.63	0.1572	1.45	0.72	2.93	0.2953

CI: confidence interval. OR: odds ratio. Nationwide primary healthcare data were used to identify the study population of women with uncomplicated cystitis. The Swedish Prescribed Drug Register was used to identify antibiotic treatment groups. The National Patient Register (inpatient data) was used to identify hospitalization due to pyelonephritis. ^1^ Fully adjusted for history of cervical cancer and parity and individual sociodemographic factors. ^2^ Cephalosporins had seven and eight cases of pyelonephritis. Other antibiotics (J01MB, J01EB, J01EC, J01ED, J01EE, and J01CR) had no cases and were not included in the table.

## Data Availability

This study made use of several national registers; owing to legal concerns, data cannot be made openly available. Further information regarding the health registries is available from the Swedish National Board of Health (Swedish: Socialstyrelsen) and Welfare: https://www.socialstyrelsen.se/en/statistics-and-data/registers/ (accessed on 22 October 2022) and Statistics Sweden (Swedish: Statistikmyndigheten, SCB): https://www.scb.se/en/services/ordering-data-and-statistics/ (accessed on 22 October 2022).

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
