# Peer review of "The Risk of Pyelonephritis Following Uncomplicated Cystitis: A Nationwide Primary Healthcare Study"

_antibiotics, 2022, doi:10.3390/antibiotics11121695_

Round 1

Reviewer 1 Report

Thank you for the opportunity to review this article. This retrospective, observational study evaluated the association of rates of pyelonephritis following an original diagnosis of uncomplicated cystitis. This study is timely as antimicrobial resistance poses an urgent threat for the medical community as highlighted by the CDC Antimicrobial Resistance Reports, and the present study may alter antimicrobial prescribing habits for one of the most frequently encountered infectious disease states, cystitis. This study evaluates risk factors, particularly antimicrobial administration, for developing pyelonephritis after uncomplicated cystitis.

I have a few general questions and comments summarized below:

Introduction:

-          Line 47: states cystitis is considered a self-limiting condition. I would consider adding verbiage to discuss asymptomatic bacteriuria, which may be misdiagnosed as cystitis. Given that inclusion for this study was ICD-10 diagnosis, I would further clarify that ASB does not require antibiotic treatment at all.

-          Line 50-52: indicates low rates of prescriptions filled by women diagnosed with cystitis. Is there any information on why that is the case or explanation of the difference in rates of filled prescriptions in the USA as mentioned in the discussion?

-          I would consider a line or two about diagnosis of cystitis (i.e. symptoms, urine analysis, and lab markers)

-          Line 71, I believe “clinic” should be the plural “clinics.”

Design and setting

-          How were patients counted if they had multiple episodes of cystitis, but no pyelonephritis recorded?

-          Even though they weren’t included, is there data on the number of patients with multiple cystitis events?

-          Line 82-83: I think this should read “Baseline occurred when a woman was first diagnosed with cystitis…”

Study population:

-          Were there any stipulations on corticosteroid dose for exclusion?

Study period:

-          Throughout, would be more consistent/definitive with wording. Ex: instead of “could be included: would say “Women were included if their first diagnosis of cystitis was between Jan 1, 2006-October 2, 2018” or something along those lines.

Statistics and sampling of study population:

-          Line 152 says women were followed for 30 or 90 days, how was it determined how long they were monitored? Should this just state they were followed for 90 days and endpoints were measured at 30 and 90 days?

Table 1:

-          Do the authors have any hypothesis on why fluoroquinolones had such a high case rate?

Discussion:

-          Line 274-275: regarding comments on hospital admission for pyelonephritis, were patients included only if they were admitted for pyelonephritis, or if there were admitted for another reason and diagnosed with pyelonephritis?

-          Line 287-288: would recommend avoiding phrases such as “we later found” just state “a study by …”

-          The limitations section is well written and summarizes the limitations well. I’m not sure there is anything to rectify with the current study, but I think these highlight the need for further considerations and study designs as retrospective studies, particularly for urinary tract infections, have a number of confounders such as misdiagnosis and overtreatment.

-          Was there any consideration given on differences in rates of pyelonephritis in patients who were diagnosed with cystitis and given a prescription that was not redeemed vs patients diagnosed with cystitis who were not prescribed antibiotics? This may be a clinically important difference in presentation at time of diagnosis/prescribing.

-          Was there any data on presentation of bacteremia or sepsis in patients given antibiotics vs not treated with antibiotics?

Supplemental tables are mentioned in the manuscript, but I do not see them included in my packet. I apologize if I missed these, but cannot seem to find them.

There was a comment in the methods section about including a sensitivity analysis. I didn't see this sensitivity analysis in the tables presented? Was this to be included in the supplemental materials? Given that this is in the main exposure in the methods, I believe more consideration should be given to commenting on this in the discussion. It may be beneficial to know if early antibiotics play a role in development of pyelonephritis after initial cystitis diagnosis. 

Author Response

PBP response to Reviewer 1

Thank you for the opportunity to review this article. This retrospective, observational study evaluated the association of rates of pyelonephritis following an original diagnosis of uncomplicated cystitis. This study is timely as antimicrobial resistance poses an urgent threat for the medical community as highlighted by the CDC Antimicrobial Resistance Reports, and the present study may alter antimicrobial prescribing habits for one of the most frequently encountered infectious disease states, cystitis. This study evaluates risk factors, particularly antimicrobial administration, for developing pyelonephritis after uncomplicated cystitis.

Response: Thank you for your positive feedback as well as for taking the time to provide useful comments when reviewing our manuscript. Please see our responses to your comments below.

I have a few general questions and comments summarized below:

Introduction:

-          Line 47: states cystitis is considered a self-limiting condition. I would consider adding verbiage to discuss asymptomatic bacteriuria, which may be misdiagnosed as cystitis. Given that inclusion for this study was ICD-10 diagnosis, I would further clarify that ASB does not require antibiotic treatment at all.

Response: Thank you for this suggestion, we have added this to the manuscript, but we feel it fitted best under the limitations section in the Discussion. The added paragraph reads:

”For example, some patients could have had asymptomatic bacteriuria, mislabelled as cystitis diagnosis, but (correctly) not treated with antibiotics…”

-          Line 50-52: indicates low rates of prescriptions filled by women diagnosed with cystitis. Is there any information on why that is the case or explanation of the difference in rates of filled prescriptions in the USA as mentioned in the discussion?

Response: Unfortunately, information to answer this question in a robust manner is, to our knowledge, not available. In Sweden, non-antibiotic treatment is a recommendation for mild to moderate cystitis and the Swedish Prescription Drug Register (utilized in this study) only have data on prescriptions picked up by patients (and not non-redeemed prescriptions). Thus, we do not know if the difference between Sweden and the U.S. is because the treating physician did not prescribe the antibiotic or because the patient did not pick up her prescription at a pharmacy (which we have discussed in the Discussion section).

-          I would consider a line or two about diagnosis of cystitis (i.e. symptoms, urine analysis, and lab markers)

Response: We have added a sentence on the symptoms and causative agents, but not the diagnostic tests as the diagnosis can be made based on symptoms alone (Bent et al. JAMA, 2002), which also is in line with the Swedish guidelines (ref 13 in the manuscript).

Bent S, Nallamothu BK, Simel DL, Fihn SD, Saint S. Does This Woman Have an Acute Uncomplicated Urinary Tract Infection? JAMA. 2002;287(20):2701–2710. doi:10.1001/jama.287.20.2701).

-          Line 71, I believe “clinic” should be the plural “clinics.”

Response: We agree, thank you. We have changed the word to “clinics”.

Design and setting

-          How were patients counted if they had multiple episodes of cystitis, but no pyelonephritis recorded?

Response: The patients were only included once, i.e., the first event of cystitis during the study period (please see paragraph 2.1. Design and Setting). We have also added a sentence in paragraph 2.3. Study period to make this clearer for the reader.

-          Even though they weren’t included, is there data on the number of patients with multiple cystitis events?

Response: We have not considered data on multiple cystitis events for this study as we are planning to conduct a separate study on recurrent cystitis in the future.

-          Line 82-83: I think this should read “Baseline occurred when a woman was first diagnosed with cystitis…”

Response: We agree and have changed the sentence accordingly.

Study population:

-          Were there any stipulations on corticosteroid dose for exclusion?

Response: In this study we did not consider the corticosteroid dose as few women (around 1%) were excluded due to corticosteroid use and we do not believe that the results would have been altered if we would have considered the dose. However, in a clinical context the dose would be important as a low dose would likely not be considered to be a complicating factor.

Study period:

-          Throughout, would be more consistent/definitive with wording. Ex: instead of “could be included: would say “Women were included if their first diagnosis of cystitis was between Jan 1, 2006-October 2, 2018” or something along those lines.

Response: Thank you for the suggestion, we have revised the sentence accordingly.

Statistics and sampling of study population:

-          Line 152 says women were followed for 30 or 90 days, how was it determined how long they were monitored? Should this just state they were followed for 90 days and endpoints were measured at 30 and 90 days?

Response: We have revised the sentence according to your suggestion.

Table 1:

-          Do the authors have any hypothesis on why fluoroquinolones had such a high case rate?

Response: Fluoroquinolones is the first-line antibiotic to pyelonephritis in Sweden (ref 13 in the manuscript) and internationally recommended as such (ref 5 in the manuscript). Thus, it is possible that some diagnoses were mislabelled (i.e., cystitis instead of pyelonephritis) and this may lie behind the high case rate. We have added a sentence on this in the discussion:

“Fluoroquinolones are recommended as first-line antibiotics for outpatient pyelonephritis in both Sweden and internationally (5, 13)."

Discussion:

-          Line 274-275: regarding comments on hospital admission for pyelonephritis, were patients included only if they were admitted for pyelonephritis, or if there were admitted for another reason and diagnosed with pyelonephritis?

Response: The hospital diagnosis of pyelonephritis included both those that were admitted for pyelonephritis and those that were admitted for another reason and diagnosed with pyelonephritis. We have clarified this in the Methodology by replacing “due to” with “with a diagnosis of”. Please see the paragraph with the subheading “2.6 Outcomes”.

-          Line 287-288: would recommend avoiding phrases such as “we later found” just state “a study by …”

Response: We have revised the sentence accordingly.

-          The limitations section is well written and summarizes the limitations well. I’m not sure there is anything to rectify with the current study, but I think these highlight the need for further considerations and study designs as retrospective studies, particularly for urinary tract infections, have a number of confounders such as misdiagnosis and overtreatment.

Response: Thank you for this comment that we agree with.

-          Was there any consideration given on differences in rates of pyelonephritis in patients who were diagnosed with cystitis and given a prescription that was not redeemed vs patients diagnosed with cystitis who were not prescribed antibiotics? This may be a clinically important difference in presentation at time of diagnosis/prescribing.

Response: This is an important comment.  Unfortunately, we do not have access to prescription data (yes/no) as the nationwide Swedish Prescription Drug Register (utilized in this study) only have data on prescriptions redeemed by patients. Therefore, no conclusion can be drawn on potential differences based on whether no antibiotic treatment was due to no prescription (healthcare) or no pick up (patient).

-          Was there any data on presentation of bacteremia or sepsis in patients given antibiotics vs not treated with antibiotics?

Response: Unfortunately, we do not have access to microbiological data. In addition, it was not the scope of the present study to assess sepsis as we are currently conducting other studies on sepsis where this outcome will be included.

Supplemental tables are mentioned in the manuscript, but I do not see them included in my packet. I apologize if I missed these, but cannot seem to find them.

Response: We believe that they were included in the original submission but we will upload them again with this revision.

There was a comment in the methods section about including a sensitivity analysis. I didn't see this sensitivity analysis in the tables presented? Was this to be included in the supplemental materials? Given that this is in the main exposure in the methods, I believe more consideration should be given to commenting on this in the discussion. It may be beneficial to know if early antibiotics play a role in development of pyelonephritis after initial cystitis diagnosis. 

Response: This is correct, we have included a sensitivity analysis in the supplementary material. We describe the analysis in the Methods section and the results in the Results section. We agree that this would also be of value to mention in the Discussion section, and it is correct that there seems to be an increased value of early antibiotic treatment for cystitis in this population. We have added this information at the end of the Discussion (just prior to the Conclusion).

Reviewer 2 Report

I appreciate the efforts and dedication about what authors have studied. The manuscript is well written and easy to be comprehended. Authors well describe the methodology and protocols systematically and provide tables and graphs in detail. The author well discusses the limitations and strengths of the study. Authors also provide a logical and clear discussion.

There are two points interesting to me.

1. Table 3, patients who received Fluoroquinolones, Cephalosporins, or Others antibiotics have significant higher OR to suffer pyelonephritis. Authors discussed these findings with “Another contributing factor may be that most of these antibiotics mainly have been recommended for outpatient treatments of pyelonephritis (both internationally (5) and in Sweden(13)). Thus, it is possible that these women already had pyelonephritis symptoms at the time of inclusion and were mislabeled with a cystitis diagnosis but prescribed the recommended outpatient treatment for pyelonephritis” How to explain this finding? Do these patients have shorter time to be diagnosed with pyelonephritis than those who received treatment other than Fluoroquinolones, Cephalosporins, or Others antibiotics?

2. According to authors, previous study (ref. 4 - Sociodemographic factors and uncomplicated pyelonephritis in women aged 15-50 years: a nationwide Swedish cohort register study (1997-2018)), originating from the Middle East/North Africa and low income were also associated with higher risks of pyelonephritis, but not living in northern Sweden. However, in the current study, “low income, and originating from the Middle East was inversely associated with pyelonephritis compared to their corresponding references, while living in northern Sweden was associated with higher odds of pyelonephritis. How to explain these differences between two studies?

Thank you.

Author Response

PBP response to Reviewer 2

I appreciate the efforts and dedication about what authors have studied. The manuscript is well written and easy to be comprehended. Authors well describe the methodology and protocols systematically and provide tables and graphs in detail. The author well discusses the limitations and strengths of the study. Authors also provide a logical and clear discussion.

Response: Thank you for your positive feedback as well as for taking the time to provide useful comments when reviewing our manuscript. Please see our responses to your comments below.

There are two points interesting to me.

  1. Table 3, patients who received Fluoroquinolones, Cephalosporins, or Others antibiotics have significant higher OR to suffer pyelonephritis. Authors discussed these findings with “Another contributing factor may be that most of these antibiotics mainly have been recommended for outpatient treatments of pyelonephritis (both internationally (5) and in Sweden(13)). Thus, it is possible that these women already had pyelonephritis symptoms at the time of inclusion and were mislabeled with a cystitis diagnosis but prescribed the recommended outpatient treatment for pyelonephritis” How to explain this finding? Do these patients have shorter time to be diagnosed with pyelonephritis than those who received treatment other than Fluoroquinolones, Cephalosporins, or Others antibiotics?

Response: Fluoroquinolones are the first-line antibiotics for pyelonephritis in Sweden, and cephalosporins (i.e. ceftibuten and cefixime) as well as trimethoprim/sulfamethoxazole (“Other antibiotics” in this paper) are also recommended for pyelonephritis in women (ref 13 in the manuscript). Thus, a possible explanation to the findings of higher odds of pyelonephritis associated with Fluoroquinolones, Cephalosporins, or Other antibiotics in this study are confounding by indication and mislabelled diagnosis (i.e. cystitis instead of pyelonephritis). However, we have no reason to believe that the patients treated with these antibiotics had a shorter time to be diagnosed with pyelonephritis than those treated with other antibiotics or no antibiotic.

  1. According to authors previous study (ref. 4 - Sociodemographic factors and uncomplicated pyelonephritis in women aged 15-50 years: a nationwide Swedish cohort register study (1997-2018)), originating from the Middle East/North Africa and low income were also associated with higher risks of pyelonephritis, but not living in northern Sweden. However, in the current study, “low income, and originating from the Middle East was inversely associated with pyelonephritis compared to their corresponding references, while living in northern Sweden was associated with higher odds of pyelonephritis. How to explain these differences between two studies?

Response: This is an interesting difference between the two studies and several reasons might lie behind these discrepancies. The two studies are based on two different study populations. Reference 4 is a study on all Swedish women 15-50 years at baseline (1997) in Sweden, exploring risk factors for pyelonephritis over a 22-year time period. The present study, on the other hand, is a study on women aged 18-65 years with cystitis and their risk of developing pyelonephritis.

Round 2

Reviewer 1 Report

Thank you for the opportunity to review this manuscript. All comments and questions were addressed appropriately and the responses are appreciated!